# EEG Dataset Collection for Mental Workload Predictions in Flight-Deck Environment

**DOI:** 10.3390/s24041174

**Published:** 2024-02-10

**Authors:** Aura Hernández-Sabaté, José Yauri, Pau Folch, Daniel Álvarez, Debora Gil

**Affiliations:** 1Computer Vision Center (CVC), C/ Sitges, Edifici O, 08193 Bellaterra, Spain; jyauri@cvc.uab.cat (J.Y.); debora@cvc.uab.cat (D.G.); 2Engineering School, Universitat Autònoma de Barcelona, C/ Sitges, Edifici Q, 08193 Bellaterra, Spain; pau.folch@uab.cat; 3Aslogic, Av. Electricitat, 1-21, 08191 Rubí, Spain; dalvarez@aslogic.es

**Keywords:** mental workload, serious games, flight simulation, EEG physiological data, deep learning, transfer learning

## Abstract

High mental workload reduces human performance and the ability to correctly carry out complex tasks. In particular, aircraft pilots enduring high mental workloads are at high risk of failure, even with catastrophic outcomes. Despite progress, there is still a lack of knowledge about the interrelationship between mental workload and brain functionality, and there is still limited data on flight-deck scenarios. Although recent emerging deep-learning (DL) methods using physiological data have presented new ways to find new physiological markers to detect and assess cognitive states, they demand large amounts of properly annotated datasets to achieve good performance. We present a new dataset of electroencephalogram (EEG) recordings specifically collected for the recognition of different levels of mental workload. The data were recorded from three experiments, where participants were induced to different levels of workload through tasks of increasing cognition demand. The first involved playing the N-back test, which combines memory recall with arithmetical skills. The second was playing Heat-the-Chair, a serious game specifically designed to emphasize and monitor subjects under controlled concurrent tasks. The third was flying in an Airbus320 simulator and solving several critical situations. The design of the dataset has been validated on three different levels: (1) correlation of the theoretical difficulty of each scenario to the self-perceived difficulty and performance of subjects; (2) significant difference in EEG temporal patterns across the theoretical difficulties and (3) usefulness for the training and evaluation of AI models.

## 1. Introduction

Abnormal cognitive states reduce human performance and diminish their ability to solve tasks. There is a wide variety of anomalous mental states that highlight mental workload, fatigue, distraction, and stress, as they decrease task performance, delay response capacity time, can block physical actions, and can lead to health and psychological disorders. Mental workload (MW or WL) has become of special interest in several areas because it affects overall human productivity and efficiency.

MW refers to the amount of mental resources required to perform a cognitive task [1]. Due to the inherent differences between subjects, MW strongly depends on each individual’s ability, psychological motivation, and the surrounding environment [2,3]. In general, the more difficult a task is, the greater the mental workload and its impact on correlated mental states [4]. For example, when a high MW is in place for a long time, fatigue appears, and stress arises [5]. On the contrary, when the MW is low for an extended period of time, the mind may become distracted and bored, which can lead to drowsiness [4]. Both cases can be harmful, especially in some activities that require a mental effort to succeed in a task [6], such as flying a plane [7] or driving a car [8], which can lead to catastrophic accidents.

The multifaceted nature of MW prevents direct evaluation, but it can be feasibly inferred from other quantifiable variables. On the one hand, a common method to evaluate MW is based on the measurement of the performance achieved during task loading, usually using questionnaires to capture the self-perceived workload of each participant. The NASA Task Load Index (NASA-TLX) [9] is among the most used questionnaires that capture the self-subject perception of a performance, complexity, time demand, and effort of a certain task. On the other hand, MW can also be evaluated using the physiological responses of the subject during the task [1]. Physiological data provides a more reliable measurement than the psychologically dependent self-subject reports. Among the great variety of physiological sensors, several studies have been carried out using a wearable electroencephalogram (EEG) due to their low cost and easy use [10,11].

Usually, MW studies focus on assessing and detecting MW in specific human communities, such as pilots during flights [1], drivers on the road [12], and other activities [13]. N-back tests and other serious specifically designed games have been proven to be useful in investigating MW since the use of working memory that they demand can provoke workload [14,15]. Although many datasets have been published using the N-back test [16,17], they are usually too short and mostly restricted to two highly differentiable mental tasks. Other researchers have collected specific datasets to study MW in specific areas, mainly in aeronautical and automotive scenarios, due to catastrophic results in aviation and car accidents. For example, in aeronautics, datasets are collected from computer-based flight simulators [18], immersive cockpit simulators [19], and real flights [20]. However, these datasets are generally private and have restricted access. Analogously, the datasets collected in the automotive industry are also too limited [21].

In this work, we present a publicly available dataset (https://doi.org/10.5565/ddd.uab.cat/259591 (accessed on 26 December 2023)) to recognize different levels of MW. It has different levels of workload, including a baseline (BL) or normal cognitive state. Part of this dataset has previously been used in our related work [22], and we claim that it can be used for research purposes to test new methods for analyzing and evaluating MW. Furthermore, this dataset has been specifically designed to enable the validation of models able to transfer knowledge to flight scenarios, which are hard and expensive to collect.

The current repository contains physiological EEG recordings from subjects facing tasks of different complexity in three different scenarios. In the first scenario, data are collected from subjects performing three variants of the N-back test to induce low, medium, and high MW. In the second scenario, data are collected from the Heat-the-Chair game, a specifically designed serious game that combines simple and simultaneous task modes, emphasizing attention and multitasking abilities. In the third scenario, data are collected in an Airbus A-320 flight simulator cockpit, in which the pilot addresses several real flight situations. The total amount of data collected is 48 sessions from 16 participants in the N-back test for a total of 34 h of recordings, 34 sessions from 17 volunteers in the Heat-the-Chair game for a total of 7 h of recordings, and 5 flight sessions with 2 pilots for a total of 95 min. In addition, the ground truth of each task, the theoretical MW complexity, the self-perceived MW complexity, the scores achieved in the games, and the NASA-TLX answers are provided.

The dataset has been validated in three aspects. First, the validity of the theoretical MW complexity has been assessed by correlating the performance obtained by the subjects with their self-perceived difficulty and game scores. Second, the quality of the WL assessment of EEG recordings has been validated by correlating their temporal patterns to the theoretical MW. Finally, the usefulness of the whole dataset for the implementation of AI systems has been assessed using the presented dataset for training and validating a DL method for the recognition of MW.

## 2. Data Acquisition Description

In this section, the main characteristics of the EEG device, participants, and experimental scenarios (design, implementation, and experiment structure) used to collect the database are described.

### 2.1. EEG Device Characteristics

The dataset described in this paper contains EEG signals recorded for a set of subjects and a set of experiments. Signals have been recorded with an Emotiv Epoc X EEG. As shown in Figure 1, it consists of a portable, wireless, high-resolution, 14-electrode EEG system, according to the International 10–20 System, that communicates via Bluetooth in real time. The electrodes are placed over the head scalp and record the electrical activity of the brain. This device provides the raw signals in μV at a sampling rate of 128 Hz. Furthermore, the sensor provides the power band for the major brain rhythms (beta: 4–8 Hz, alpha: 8–12 Hz, beta low: 12–18 Hz, beta high: 18–25 Hz and Gamma: >25 Hz). Emotiv gives 8 power samples per second computed over the previous 2 s.

Signals are recorded directly from the headset and undergo significant signal processing and filtering to remove mains noise and harmonic frequencies. In particular, signals are sampled at 2048 Hz, a dual-notch filter is applied at 50 Hz and 60 Hz, and a low-pass filter is computed at 64 Hz. Finally, data are filtered down to 128 Hz or 256 Hz for transmission.

### 2.2. Participants

For all the experiments, written consent was obtained from each participant. The consent form explains the goal of the experiment and describes what kind of data are collected and the terms of privacy in the use of personal data. Additionally, it emphasizes that the data released to the general public does not contain information that can directly identify the subject and that any data and research results already shared with other investigators or the general public cannot be destroyed, withdrawn, or recalled. Each consent was hand-signed by each subject on the day of the first experiment.

The participants in all experiments were healthy people without any condition that might have caused an imbalance in the recorded data. The characteristics of all the participants are detailed as follows:The N-back test experiment: 16 subjects (all male), with ages ranging from 20 to 60 years, participated in the experiments. The volunteers belonged to three different university research centers and shared a scientific background with different levels of expertise (students, junior researchers, senior researchers, or professors).The Heat-the-Chair game experiment: 17 subjects (12 male and 5 female), with ages ranging from 20 to 60 years, participated in the experiment. The volunteers shared the same characteristics as participants in the previous experiment, and seven of them completed the preceding test.The flight simulation experiment: two professional pilots, but with different experience levels, participated in all flight missions, but they exchanged roles depending on the mission. Table 1 details the information of the pilots.

Figure 2 illustrates some of the participants in the different experiments. All of them signed a written consent to publish their images.

### 2.3. N-Back Test Experiment

We used the N-back test game to induce different levels of mental workload in participants. This type of experiment requires the ability to manage one or two N-back tasks simultaneously, taking into account the insights shown in the n-trial before, so it demands a high usage of memory to complete the tasks. In particular, we designed three experiments with different levels of complexity (low, medium, and high), and each subject performed all the experiments, randomly assigned, using a computer. The three variants of the N-back test to induce mental workload were implemented as follows:Low mental workload—position 1-back: As Figure 3 shows, a square appears every few seconds in one of nine different positions on a regular 3×3 grid over the screen. The player must press a key on the keyboard when the position of the square on the current screen is the same as the position of the square that appeared on the previous screen.Medium mental workload—arithmetic 1-back: As Figure 4 shows, an integer number between 0 and 9 appears every few seconds on the screen, while an arithmetic operation (plus, minus, times, and divide) is audibly presented. The player must solve this operation using the number that appeared on the previous screen and the current one. Results must be typed using the numerical keys.High mental workload—dual position and arithmetic 2-back: This test combines the two previous ones. As Figure 5 shows, an integer number between 0 and 9 appears every few seconds in one of nine different positions on a regular grid. At the same time, an operator is presented visually. As before, players must type the solution of this operation using the number that appeared on the two screens before and the current one. In addition, players must press a key in case the position of the current number is the same as the position of the number shown two screens before.

Experiment Structure. Before playing and recording the data, the subject was informed about the rules and trained in the game for five minutes. For training, the dual position and audio 1-back mode were used, which simultaneously combines position and audio, taking into account the 1-back step, i.e., a number between 0 and 9 is audibly presented, and the player must press a key if it matches the one emitted in the previous screen, another one if its position matches, and another one if both matches occur.

We assume that, in the absence of any required mental effort, subjects will exhibit a baseline mental workload, and their physiological responses will accordingly be at a minimum scale. Additionally, we expect that the baseline levels will vary among individuals. To induce this baseline state, participants watched a relaxing video for 10 min before engaging in the N-back tests. Subsequently, they played the game for 20 min. After completing the game, they responded to the NASA-TLX questionnaire [9] to provide their subjective perceptions of the mental workload and effort demanded by the game. Finally, to come back to a calm state after the task, subjects underwent a 10-minute recovery step, mirroring the baseline stage. The experimental protocol is described in Figure 6. During each session, all neuro-physiological responses were continuously recorded. The dataset only contains signals from the baseline, the game task, and the recovery phase, removing the parts not strictly belonging to the experiment. The dataset also contains the results from the NASA-TLX questionnaire and the achieved scores of the player, which correspond to the number of hits.

### 2.4. Heat-the-Chair Experiment

This game was specifically designed to create a scenario in which simultaneous tasks must be performed, replicating the demand for concentration and alertness of pilots while flying. The game consists of completing as many objectives as possible in 10 min. Completing an objective consists of obtaining and using the necessary pieces to form a 4-digit number, which appears at the top left of the screen for 10 s and then disappears, reappearing for 5 s every 1 min while the objective is not achieved. Once the correct pieces have been obtained and the target puzzle has been completed (the 4-digit number), the player increases the punctuation, and a new target number to be completed appears automatically. Figure 7 shows the game user interface. The target number appears in the upper left panel, while the pieces that the player obtains are in the lower right panel. Notice that the bottom row is designated for storing the rewarded pieces (in cyan), while the top row is dedicated to dragging and dropping the pieces to replicate the 4-digit number.

To obtain pieces, the player must perform two main tasks:Bars with sliders: As we can observe in Figure 7, there are two colored bars in the bottom central-hand panel with sliders that move in the horizontal and vertical directions. The player must keep the sliders in the center of the bars using the directional keys of the keyboard.Dots: In the same panel, there is a large square that will be filled with dots. To avoid this, the player must drag them to the dashed-line box shown in the center.

In the top central panel, there is a circular button with a depleting energy bar below it. The difficulty of tasks will increase proportionally to the depletion of the energy bar: the emptier the bar, the more challenging the game will become. Thus, the player must regularly recharge the power bar using the circular button.

The key point is that the game supports two modes of operation: with or without interruptions. Interruptions are introduced in the game design to emulate the interruptions that pilots receive when interacting with Air Traffic Control (ATC). In this mode, incoming events randomly appear to be solved. In particular, five different tasks, in random order, are required to be completed by the player. Tasks can be either to report a current flight parameter (altitude, wind speed, wind direction, and bearing) or to change the number of the switch box (the switch box starts randomly at each game). Flight information is shown on the left-center portion of the screen, and the switch box is shown on the top-right portion of the screen. When an interruption arrives, an alert of messaging is shown on the bottom-left portion of the screen. The player must click and read the message. Each required task has a starting and an ending time to be completed, beyond which the player is penalized. Figure 8 depicts an interruption asking for a change to the current switch box. The start and end times to complete the task are highlighted in green and red, respectively. If the player does not complete the task or inserts incorrect information as an answer, one rewarded piece is lost.

***Experiment Structure.*** Before playing and without recording data, the subject was informed about the rules and trained in the Heat-the-Chair game without interruptions for 5 min to familiarize themselves with controls. The game mode was chosen randomly before starting the experiment.

Given that each subject randomly faces the two modes of the game, each game is recorded in separate sessions. As Figure 9 shows, a session consists of three phases. The baseline lasts 3 min, in which the subject drags balls that randomly appear on the screen and drops them to the dashed square in the center. The game has the subject play the randomly selected game mode for 10 min, either with interruptions or without interruptions. In addition, finally, there is the NASA-TLX questionnaire, which the subject fills out, indicating his/her subjective perceived game complexity. Neurophysiological responses are continuously recorded for the entire session using the Emotiv. The dataset provides the signals from the baseline and the task phase, removing the time intervals that do not take part in the experiment. The dataset also contains the player’s achieved scores and the results from the NASA-TLX questionnaire.

### 2.5. Flight Simulation Experiment

The goal of this experiment was to collect experimental data useful for quantifying the impact of an increase in the mental workload of pilots during the performance of routine flight tasks (i.e., within reference parameters) and when they must manage additional unexpected phenomena, such as wind shears, machine failures, equipment warnings, and unusual traffic. In these situations, interaction between the crew and the ATC increases, and pilots are more likely to make mistakes due to the mental workload. For that, five flight-simulated scenarios were designed to evaluate the pilots’ task load changes while they solve unexpected situations. Each flight scenario is an experiment, and each experiment is unique, with its own characteristics. The flight simulation was carried out in an immersive Airbus-320 cockpit simulator, and the chosen flight route was from Barcelona to Lleida in Spain, with an approximate duration of 14 minutes. Figure 10 illustrates the route followed by the pilot. Weather, weight, and speed conditions were fixed for all flights.

An expert flying pilot defined five flying scenarios with different levels of complexity and events:Flight 1 [easy difficulty]: The pilot performs a standard flight to be used as reference parameters.Flight 2 [medium difficulty]: During the flight, the ATC reports much traffic, so the pilot is asked to change the position of the airplane above the glide slope at high speed.Flight 3 [hard difficulty]: During the final stage of the flight, the airplane is hard destabilized by a strong wind shear, so the pilot must maneuver, recover the plane stability, and land it.Flight 4 [medium-hard difficulty]: During the flight, a malfunction during the approach provokes an engine failure that increases the crew workload.Flight 5 [medium difficulty]: This flight is similar to Flight 2 with a little variation.

Two flying pilots participated in the experiment. Before starting the experiment, pilots received a printed description of the assigned flight mission, and one was chosen as the pilot, whereas the other remained as the copilot/observer. A third pilot was monitoring the whole flight for the annotation of the pilot’s perceived complexity in time stamps of 30 s. Table 2 reports the distribution of the roles of the two flying pilots for the five flying scenarios.

Experiment Structure. The experimental protocol of each flight was divided into two phases. First was the baseline phase, in which the pilot stays on the runway awaiting the order to take off. Second was the flight phase, which at the same time can be split into three stages: the takeoff, when the flight starts, and the plane climbs; the task phase itself; and a short time for landing the plane on the ground. The task phase encompasses cruise, descent, and approach tasks, along with standard communication with the ATC, and includes the specific tasks requested for each flight simulation. Figure 11 shows the timeline of the flight simulation experiment. The dataset contains the neurophysiological responses from all phases of the experiment. It also contains the difficulty perceived by pilots obtained from a modification of the NASA-TLX questionnaire to obtain a dynamic perceived difficulty across the flight phase collected by the third pilot named above. The level of difficulty was graded from 0 to 3, with 0 being the easiest one.

## 3. Data Format and Structure

The dataset described in this paper has been made publicly available on the Digital Document Deposit of the Universitat Autònoma de Barcelona, accessible at https://doi.org/10.5565/ddd.uab.cat/259591 (accessed on 26 December 2023). No registration is required.

The dataset is provided in a compressed file workload\_dataset.zip. After decompression, the dataset contains three main folders that store the collected data for the N-back test data (folder data_n_back_test), the Heat-the-Chair data (folder data_heat_the_chair), and the flight simulation data (folder data_flight_simulator), respectively.

The next subsections explain the data format and structure for each experiment.

### 3.1. N-Back Test

EEG signals, game performance, and TLX questionnaires were stored in 3 .parquet files containing the data obtained from all participants in the following structure.

The acquisition software saves the raw data in a CSV file that has 139 columns with datetime, timestamps, the 14-electrode data, the five frequency power band data, and quality metrics supplied by the sensor. Since a session itself is split into phases, the recording is a continuous signal. We used manual annotations of the starting and ending times of each phase to remove data outside the phases. Also, all the annotations were synchronized by means of a specifically designed application. The data recorded for the session phases and all participants was stored in a .parquet file that included metadata added as three additional columns: subject, test, phase. The subject column is the identifier of the volunteer. Participants are labeled as subject_xx, for xx∈{1,…,16} a number identifying the subject. The test column identifies the variant of the N-back test (1 for the position 1-back, 2 for the arithmetic 1-back, and 3 for the dual position and arithmetic 2-back). The phase column identifies the phase in the session (1 for the baseline, 2 for the task, and 3 for the recovery).

The dataset also provides the performance of the subject’s game during the N-back test. Each subject has three measurements—one for each task difficulty. Since the task phase lasts 20 min and the trial of the game is almost two minutes, the subject played the game many times, so scores are provided as a list of punctuation. The .parquet file also includes the fields subject and test defined as before for the identification of the subject and N-back test variant.

The answers to the TLX questionnaires were also collected for all games and participants and stored in a .parquet file including the fields subject and test.

The directory tree for the dataset of the N-back test is the following:

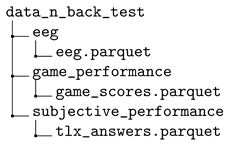

where the file eeg.parquet stores the EEG signals for all participants, the file game_scores.parquet the game scores per subject and tlx_answers.parquet their answers to the TLXs questionnaires.

### 3.2. Heat-the-Chair Game

The EEG raw data stored in CSV files for each session was processed to remove parts outside the baseline and the task phase. The data of all participants was stored in a single parquet file, including metadata for the identification of the subject, game type, and phase. The dataset also provides the performance of the subject during the game, and the self-perceived workload for each task reported in the TLXs. The directory tree for the dataset for the Heat-the-Chair game is the following: 

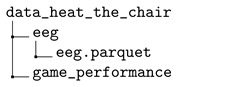


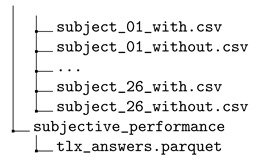



Each of the folders contains the following information:The folder eeg contains the file eeg.parquet storing a pandas dataframe with all the EEG data and three extra columns (subject, test, phase) of metadata. The field subject identifies the volunteer as ‘subject_xx_’, for ‘xx’ a two-digit number. The field test identifies the game mode: 1 for a game without interruptions and 2 for a game with interruptions. The field phase identifies the stage of the experiment: 1 for the baseline and 2 for the game.The folder game_performance contains the game scores for each subject and game into separated CSV. The name of these files follows the pattern ‘subject_xx_mode’, where xx is a two-digit number identifying the volunteer, and mode is the game type: ‘with’ for a game with interruptions and ‘without’, otherwise.The folder subjective_performance contains the file tlx_answers.parquet with the answers to TLX questionnaires for all participants and games.

From the set of 17 volunteers, the seven subjects compressed between 1 and 16 subjects have also participated in the N-back test; the rest of them, from 17 to 26, were new participants.

### 3.3. Flight Simulator

The data recorded from the flight simulator is in the folder data_flight_simulator. To make the processing of data easier, the original ‘csv’ files provided by the sensors were prepared by adding additional columns and saved into a single one, ‘parquet’. Five columns were added for both EEG and ECG. The column subject identifies the pilot who is flying (number 1 identifies pilot 1, whereas number 2 identifies pilot 2). The column flight indicates the flight experiment performed, ranging from 1 to 5. The column phase indicates the stage of the flight. Values are ‘baseline’ and ‘flight’ (see Figure 11). The column theoretical difficulty represents the expected theoretical workload induced in the pilot, and the values range from −1 to 4 to indicate easy to hard. Each flight has its own theoretical difficulty. Finally, the perceived difficulty columns provide the perceived difficulty reported by the pilot about the complexity of the assigned mission. During the simulation, at every certain interval, the pilot scores the complexity of the scenario from 0 to 3. We encoded the perceived difficulty of the ‘baseline’ stage as −1.

Finally, the flight simulator data are organized into three sub-folders:

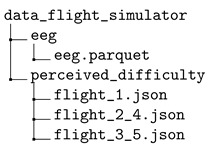



Each of the folders contains the following information:The folder eeg contains a single file eeg.parquet with the EEG data.The folder perceived_difficulty contains the perceived difficulty of the pilots during the flight. It contains three ‘.json’ files: flight1.json, flight2_4.json, and flight3_5.json.

The flight simulator experiment Wasim contains the collected data from two professional pilots who performed five simulation flights in an A320 cockpit. The expected induced degree of workload and the self-perceived workload reported by the pilots themselves are also registered.

## 4. Data Validation

In this section, we present the technical validation of the proposed data. To show its usefulness for training and evaluation of AI models, we have conducted 3 different experiments:Correlation to the Perceived Difficulty. The goal of this experiment was to assess that the theoretical difficulty of each experimental scenario can be used as a valid annotation defining a ground truth for AI models. This assessment checks that the evolution of the difficulty perceived by participants increases along with the theoretical one. For the two games, we also include the performance to check if it is decreasing as theoretical performance increases.Analysis of Differences in Temporal Patterns. The goal of this experiment was to show that the temporal waveforms of signals recorded under different WL conditions had different patterns. In particular, we have analyzed the number of spikes extracted from EEG recordings to assess whether their distribution is different across increasing levels of WL (N-back-test dataset), number of interruptions (serious game), and flight difficulty.Usefulness for Training AI models. The goal of this experiment was to assess the usefulness of the presented dataset by training a DL model using the N-back test data and testing it on the Heat-the-Chair and flight simulator data to show its transfer task capability.

In the next sections, we report the experimental setup and results obtained for each of the experiments.

### 4.1. Correlation to the Perceived Difficulty

To evaluate the technical quality of the collected data in the N-back test, we analyzed the answers to TLX questionnaires. Since TLX reports the self-perceived degree of workload enforced by the tasks, we put them in correspondence with the performance of the players. Figure 12 shows boxplots of the perceived difficulty given by the TLX questionnaire and game performance given by the percentage of correct operations for the 3 levels of difficulty of the game. On the one hand, the performance of subjects is decreasing with the theoretical difficulty of the game, as expected. On the other hand, the perceived workload of participants also has the expected increasing correlation with the theoretical difficulty. Finally, the perceived workload increases as the performance decreases, which is also consistent with the hypothesis that each test offers a challenge according to its difficulty level.

For the validation of the Heat-the-Chair game, we observe that, unlike the N-back test single memory task, this game includes simultaneous tasks (memory, perception, manual operations, and decision making) triggered by interruptions. Thus, the validation of these datasets is based on the correlation between perceived workload, participant performance, and the two modes of operation: with or without interruptions. As before, the perceived workload is given by the TLX questionnaire, while performance is given by the average time (in seconds) required to obtain pieces during the game. In this case, as the metric chosen to measure performance is the average time it takes the subject to obtain a piece, an increase in this time reflects a decrease in performance. Figure 13 shows boxplots of the perceived workload and performance obtained for games with and without interruptions. Since, in this case, the two quantities have different ranges, we show their boxplots with a standardized y-axis. The y-axis range shown in red on the left-hand side corresponds to the time needed to obtain pieces, while the range shown in blue on the right-hand side corresponds to the range of punctuation for TLX results, scaled between 0 and 50. As expected, both quantities are clearly increasing with the number of interruptions.

Given that we do not have any metric of pilot performance, for the validation of the flight simulations, we have analyzed the correlation between the distribution of the perceived difficulty (given by the dynamic TLX collected during the flight) and the global difficulty of the different flying scenarios. We recall that in this case, the perceived difficulty rates the complexity of the flight at time stamps according to 0—low, 1—mid–low, 2—mid–high, and 3—high. Figure 14 shows a boxplot of the perceived workload for the 5 flying scenarios with the median line in bold for better visualization. The theoretically easiest Flight 1 also has the lowest perceived difficulty, with values below 2 and 50% of the time stamps rated with mid–low difficulty. The twin flights, Flight 2 and Flight 5, have an identical distribution of values, with 50% of the flying time considered to be mid–low difficulty and only 25% rated as high. Flight 3, with the highest theoretical complexity, is also the one with the highest values of perceived WL, with 50% of the flight considered high-complexity by the monitoring pilot. Finally, Flight 4, of medium theoretical difficulty but with an unexpected event, is perceived as being of low–mid difficulty 50% of the time but with a 25% of high perceived complexity. This peak of complexity coincides with the time of the unexpected malfunction. This match between perceived and theoretical complexities validates as ground truth annotations both the global theoretical difficulty as well as the perceived complexity annotated by experts along the flying time.

### 4.2. Analysis of Differences in Temporal Patterns

To analyze the differences in the temporal patterns of EEG recorded at different levels of task complexity, we have decomposed the signal of each EEG node into their power spectra waves (θ, α, β, and γ). These signals are obtained by filtering each sensor signal at the following 4 frequency bands clinically related to main brain processes [11]:θ (4–8 Hz). The θ activity is seen in drowsiness, arousal, and often during meditation. Dominant θ activity is associated with relaxed, meditative, and creative states, memory recall, and ‘flow’ states. It is reported that an increase in theta, particularly frontal theta, is often associated with an increase in working memory load, especially in single-task contexts.α (8–12 Hz). The α waves are the default ‘relaxed and alert’ mode of the brain. High α levels appear in the frontal lobes during relaxation and are suppressed when other activities (like linguistic, abstract spatial thinking, or muscular) take place. It is reported that the alpha power during high-workload tasks might be lower than the alpha power during low-workload conditions.β (12–25 Hz). This band is often associated with active, task-oriented, busy, or anxious thinking and active concentration. It is reported that beta power during a high-workload task is moderately greater than beta power in low-workload conditions. Numerous studies have established the involvement of this frequency in a variety of cognitive processes such as working memory [23], language processing [24], and decision making [25]. Since the Emotiv API provides access to two sub-bands in the β zone (12–18 Hz, labeled $\beta_{l}$ and 18–25 Hz, labeled $\beta_{h}$), we have analyzed both.γ (greater than 25 Hz). The γ band activates when different populations of neurons network together to carry out demanding cognitive or motor functions requiring fast. Coupled processing is required [26]. It is reported that γ activation is related to emotions, perception, and attention. However, there are no conclusive studies of any relationship between γ power and WL.

For each power band, we computed the peaks of each node waveform as an indicator of temporal variability in brain activity associated with WL. Peaks were computed as local maxima with a value above 95% percentile of the node power spectra band.

Figure 15, Figure 16 and Figure 17 show the barplots for the average number of peaks of the recordings for all participants of a given experiment. We show a different barplot for each band (rows) and experimental scenario (columns). For each barplot, we have grouped by node the number of peaks obtained under the different complexities of the experimental scenarios. For the N-back test and serious game, we have used the global theoretical difficulty, while for the simulator, given the variety of complexity along a single flight, we have used the perceived workload grouped into easy-mild (scores 0, 1) and mild-high (scores 2, 3). The expected pattern is that the number of peaks is a monotonous function of the level of complexity for some of the EEG nodes.

For the N-back test, the peaks of the θ band are increasing in game complexity for all sensors except the occipital O7. This is aligned with the fact that the N-back test is a memory-demanding activity. The increasing pattern is also observed in most of the left sensors of the α band associated with spatial thinking. The β and γ bands do not present a monotonous pattern, which is not surprising given that they are not associated with memory items. For the Heat-the-Chair game, θ and α are again the bands with a more prominent increasing pattern, which is followed for all nodes. This reflects the increased complexity in memory items and spatial coordination required to play the game. In this case, the β and γ bands also have an increasing pattern for most nodes, which indicates that the game requires multitask solving (γ) and provokes anxious thinking (β). For the flight simulator, the distribution of peaks is different. The θ wave has a small decreasing pattern, although with very similar values (see ranges in Table 3) which are the highest ones. This indicates that flying is a knowledge-based task that requires recalling learned concepts. The peaks of the α wave do not follow a well-defined pattern and are missing for some sensors. In fact, it is the band that has the lowest values (see ranges in Table 3). This suppression of the α band indicates that even for easy flights, pilots flying an aircraft is a highly demanding mental task. The $beta_{h}$ band is the only one presenting a clear increasing pattern for almost all nodes, which can be attributed to the fact that during hard flying conditions, the pilot needs to make a lot of quick decisions. This is also reflected in the increasing pattern of the γ wave for both parietal nodes.

Table 3, Table 4 and Table 5 report descriptive statistics for the number of peaks summarized as mean ± standard deviation computed for all nodes. The expanded descriptive statistics for each sensor can be found in Appendix A. For the N-back test, the ranges of all bands are higher during the tests compared to the ones of the baseline. However, the only band that has increasing ranges along test complexity is θ. For serious games, the ranges of all bands are higher in the games with interruptions. Finally, for the simulator, only $\beta_{h}$ and γ bands have higher ranges for mild-hard phases of the flight.

To detect if the differences in ranges were significant, we adjusted a generalized regression model [27] for the number of peaks with the complexity as a fixed factor and the sensor as a random effect to account for differences between them and correct for repeated measures. A different model was adjusted for each power band and experimental scenario. For the detection of significant differences, all models compare the number of peaks of each complexity level with the lowest-complexity ones. For each regression model, we report model parameters, *p*-values for significance in fixed effects, 95% CI for their mean values. In the case of a change in scale or transformation of data required to satisfy model assumptions, the CIs were back-transformed to the original scale. For all statistical analysis, a *p*-value <0.05 was considered significant. Statistical analyses were conducted using R version 4.3.2.

Table 6, Table 7 and Table 8 report a summary of the regression models for each of the experimental scenarios. For the N-back test, the complexity factor was significant for all bands except γ. For the θ band, peaks were significantly increasing with complexity. This is not the case for the remaining bands, where models detect a significant decrease in the number of peaks of the Medium complexity. The peaks of the high-complexity waves are significantly higher than the low-complexity waves, except for γ. The models for the Heat-the-Chair game detect a significant increase in the number of peaks of games with interruptions for all bands. Finally, for the flight simulator, models detect a significant increase in the number of peaks of the β waves and a significant decrease for θ. There are no significant differences for α and θ.

### 4.3. Usefulness for Training AI Models

In this section, we illustrate the usefulness of the EEG dataset by training AI models. We report partial results of previous work on DL models for WL detection, published in [22]. In that work, several architectures for EEG channel fusion were presented and validated on the n_back test data using a leave-one-subject-out cross-validation scheme. The best performers trained on the whole n_back set were also validated on the Heat-the-Chair set to assess the task transfer of models. We summarize the main findings and report preliminary results obtained on the flight simulator by models trained on the nback test.

Our best-performing architecture was a convolutional neural network (CNN) that fuses the channels of the features obtained after the convolutional block and before the Fully Connected (FC) layers. Concretely, signals are processed as follows: input data are taken by the input data module and feed the convolutional module, which performs feature extraction. The number of channels remains along convolutions so that a channel fusion unit transforms them into a single signal channel for each feature. The outcome of the previous unit is flattened and processed by two FC layers to combine features and perform output predictions.

Figure 18 depicts the proposed neural architecture. Notice that *C* corresponds to the number of channels, *T* to the time window, *L* to the convolution, and $n_{f}eats$ to the number of features extracted by the convolutional block $L^{th}$. Hence, after the convolutional process *L*, $x^{C\times T}$ becomes $x^{n\_\mathrm{feats}@C\times (T//L)}$, and T//L is due to the pooling operation in each convolutional block. During the channel fusion unit, the input $x^{n\_\mathrm{feats}@C\times (T//L)}$ becomes $x^{n\_\mathrm{feats}@1\times (T//L)}$. The convolution unit has 3 blocks consisting of one convolutional layer with max pooling and with 16, 32, and 64 neurons for each convolutional layer, respectively. The classification layer has 256 neurons. The output unit has 2 blocks consisting of one convolutional layer before the classification layer. The first one has 64 neurons, and the second one projects convolutional features also using 64 neurons.

For the validation of the models, the N-back test data were split using a leave-one-subject-out scheme, and 16 independent models were trained on 15 participants and tested in the remaining one. The models were trained for the classification of BL and medium phases using a weighted cross-entropy loss to compensate for imbalances between baseline and workload phases. We used a batch size of 750, Adam as the optimization method, 100 epochs, and a learning rate of 0.0001. The quality metrics were the sensitivity and specificity, considering BL the positive class.

Results show a 76.25% sensitivity and 87.81% specificity in WL detection for a leave-one-out subject evaluation in the N-back-test data and good task transfer with the detected WL increasing with the number of interruptions in the Heat-the-Chair game (see Figure 19).

Regarding the capability of task transfer, Figure 20 shows the barplots for the number of predictions in BL and WL for 4 of the 5 flights (the last has not been checked, as it is very similar to the second). As expected, the highest number of BL detections is for Flight 1, while Flight 2 and Flight 4 show more WL detections since they are characterized as more difficult. Flight 3 was not as expected and could be attributed to the discrepancies in waveform between synthetic games and the simulator, detected in Section 4.2.

## 5. Conclusions

This paper provides a complete dataset of physiological recordings from electroencephalogram (EEG) and electrocardiogram (ECG) devices, which are useful for testing methods that recognize mental workload. Three different experiments have been presented in whose participants were induced to different levels of workload:on the well-known dual N-back gameon a specifically designed serious game mimicking the increase of workload an aircraft pilot can sufferon a flight simulator

Technical validation at three different levels shows the correlation between objective measures from the experiments and the corresponding subjective self-perceived complexity from subjects. Our games are specifically designed to target mental activities and, thus, can be used to assess the capability of a physiological sensor to detect mental WL or any other specific mental effort. Moreover, we have shown that they could be powerful means for collecting unambiguous annotated data valid for training AI models.

However, there is room for improvement. The results obtained on the transfer to the simulator are sub-optimal for an AI system deployable in the cockpit. The analysis of the power band patterns of Section 4.2 shows that the synthetic games have a mental demand different from that of flying pilots. Therefore, more specific serious games should be designed to guarantee a fully successful transfer to flight-deck scenarios.

## Figures and Tables

**Figure 1 sensors-24-01174-f001:**
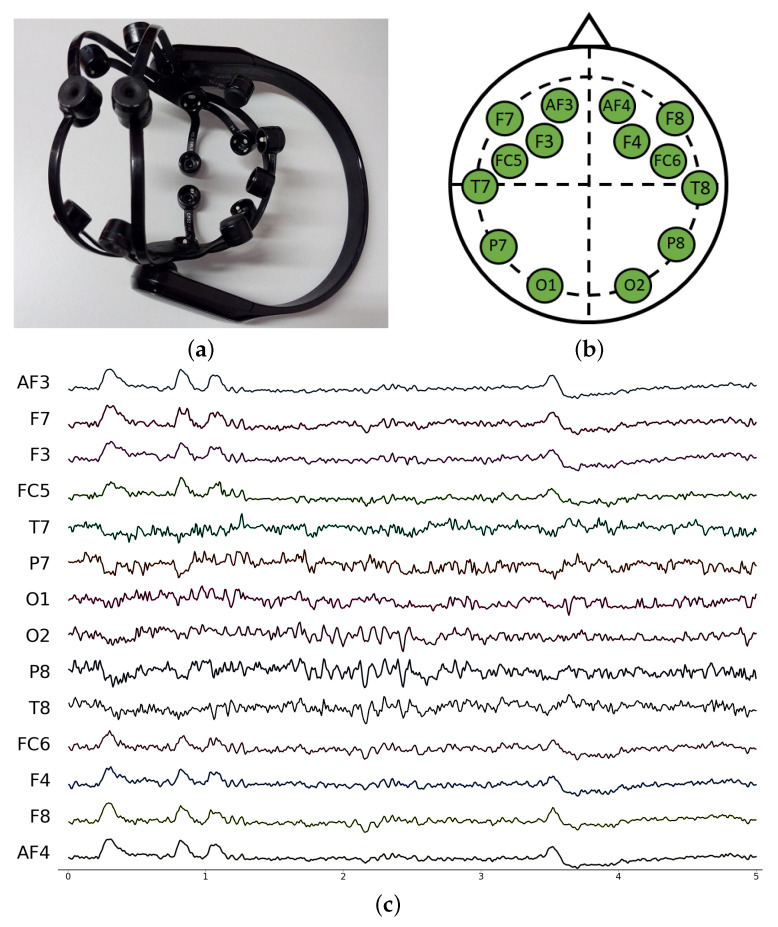
EEG Emotiv Epoc X headset used for recording the data (**a**), spatial distribution of sensors for the EEG Emotiv Epoc X (**b**), signal pattern recorded during 5 s (**c**).

**Figure 2 sensors-24-01174-f002:**
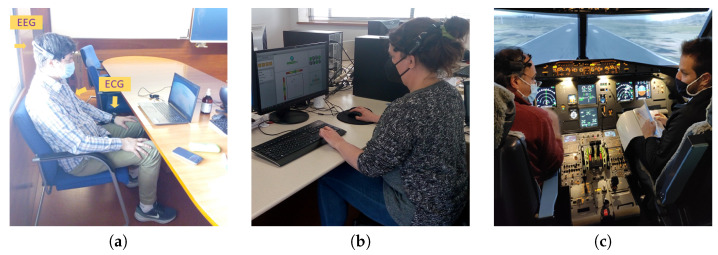
Volunteers, during the experiments, performing: (**a**) N-back test, (**b**) Heat-the-Chair game and (**c**) flight simulation in the cockpit of an A320.

**Figure 3 sensors-24-01174-f003:**
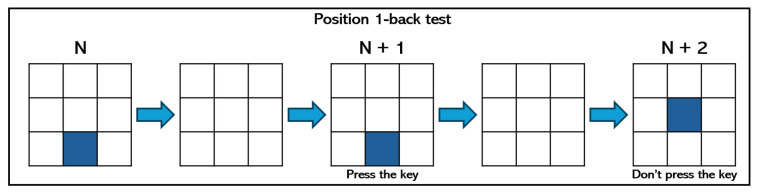
Example of position 1-back test.

**Figure 4 sensors-24-01174-f004:**
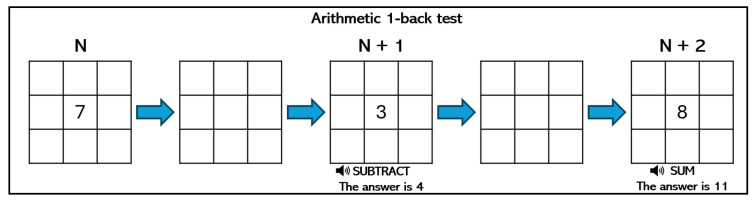
Example of arithmetic 1-back test.

**Figure 5 sensors-24-01174-f005:**
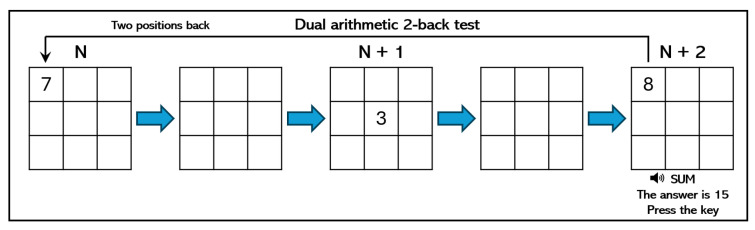
Example of dual position and arithmetic 2-back test.

**Figure 6 sensors-24-01174-f006:**
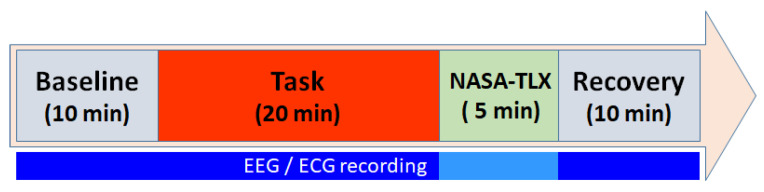
Timeline of the N-back test experimental protocol.

**Figure 7 sensors-24-01174-f007:**
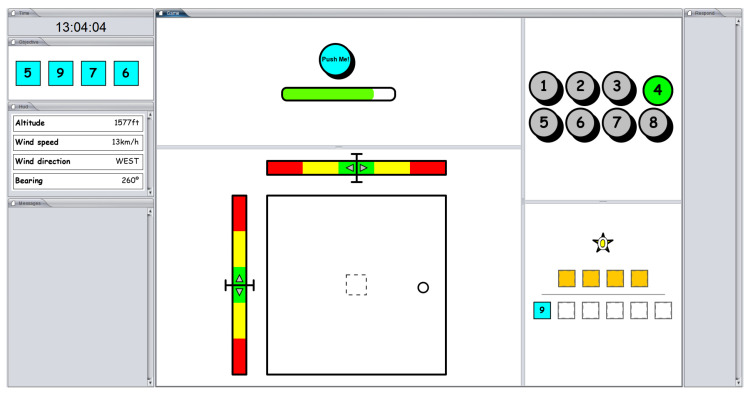
The Heat-the-Chair game user interface.

**Figure 8 sensors-24-01174-f008:**
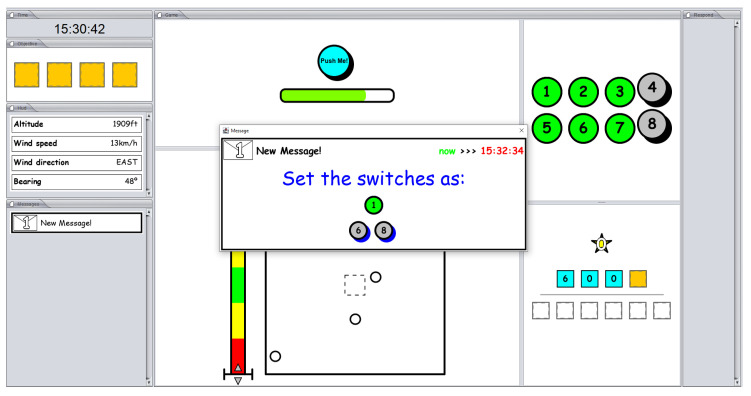
The Heat-the-Chair game with an interruption message.

**Figure 9 sensors-24-01174-f009:**
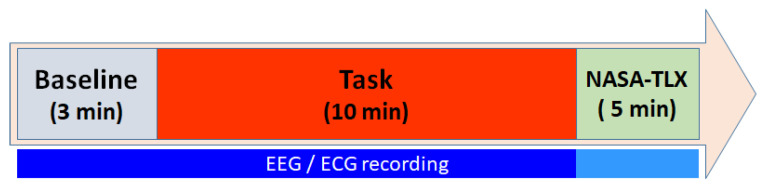
Timeline of the Heat-the-Chair experimental protocol.

**Figure 10 sensors-24-01174-f010:**
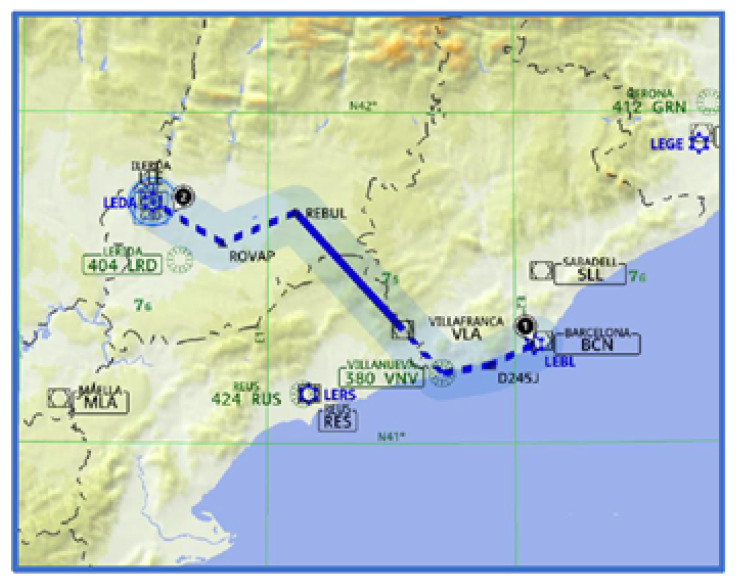
Flight simulation route.

**Figure 11 sensors-24-01174-f011:**
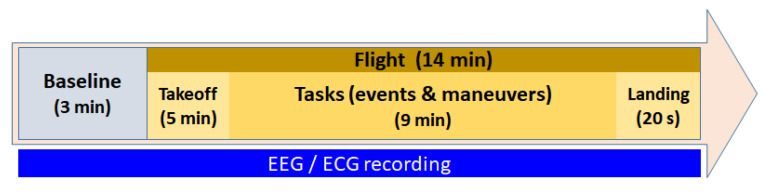
Timeline of the flight simulation experimental protocol.

**Figure 12 sensors-24-01174-f012:**
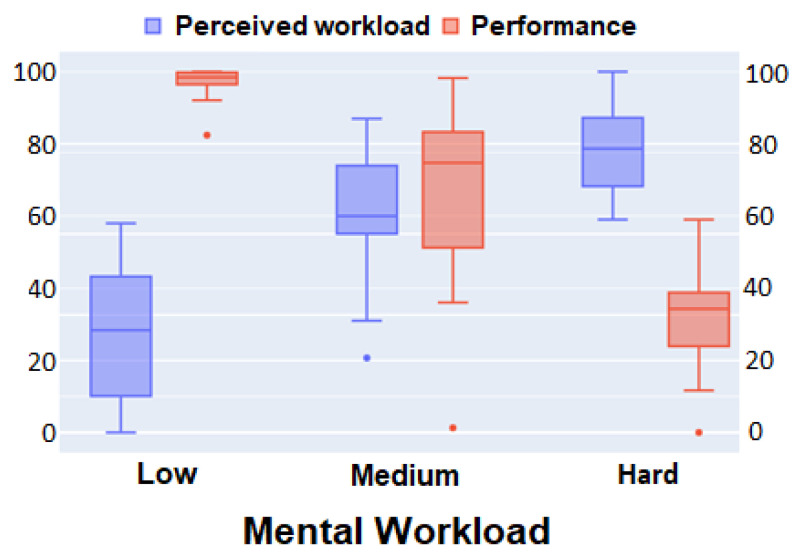
TLX analysis in the N-back test. Perceived workload and game performance across theoretical difficulty.

**Figure 13 sensors-24-01174-f013:**
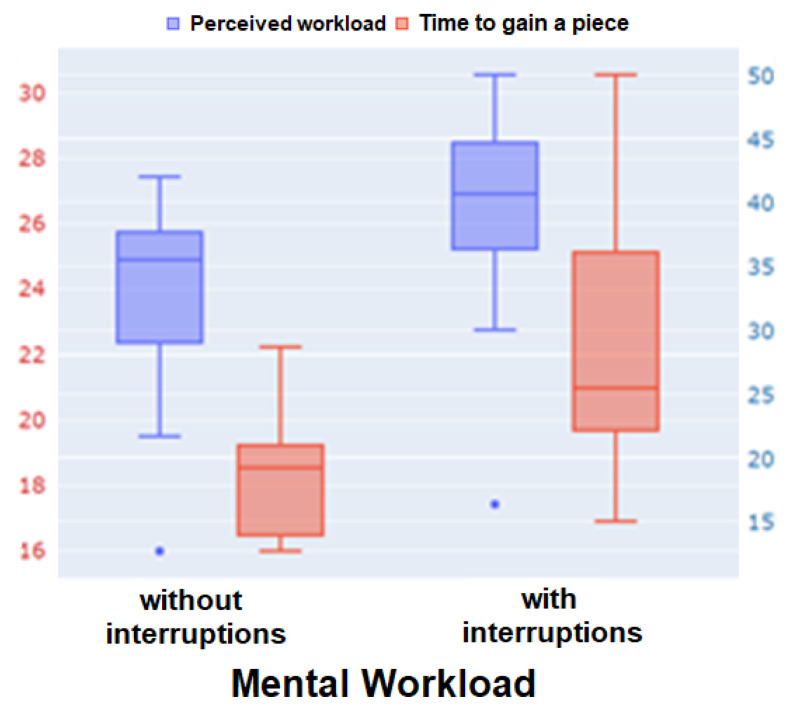
TLX analysis in the Heat-the-Chair test. Perceived workload and game performance across interruptions.

**Figure 14 sensors-24-01174-f014:**
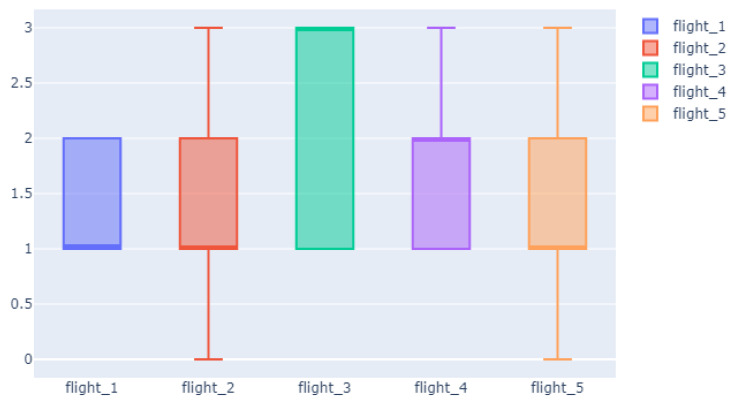
TLX analysis in Simulated flights. The perceived workload in the flight scenarios.

**Figure 15 sensors-24-01174-f015:**
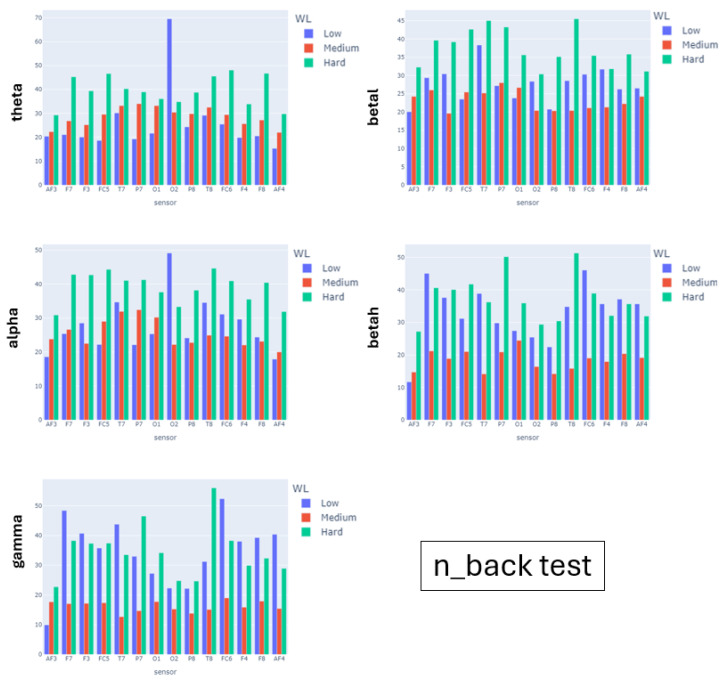
Number of Peaks of EEG Nodes Power Bands. Barplots for the n_back test Experiment.

**Figure 16 sensors-24-01174-f016:**
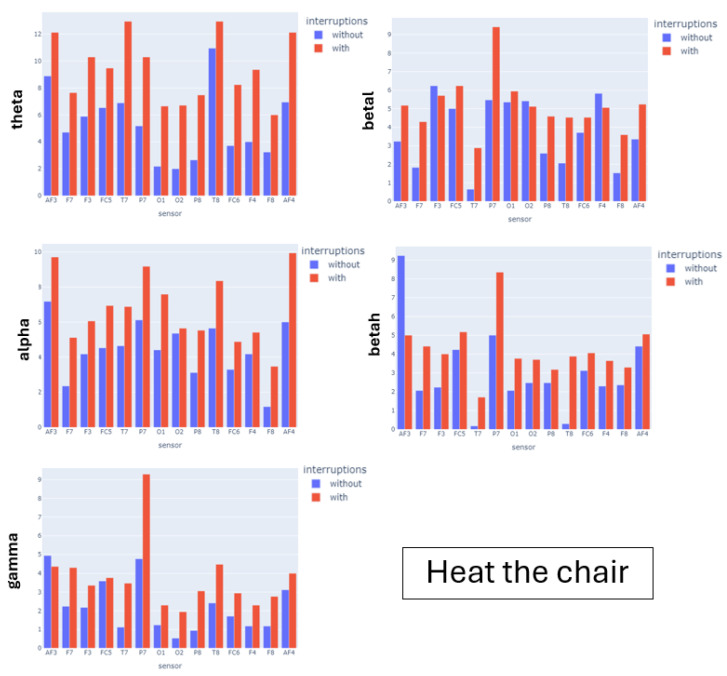
Number of peaks of EEG node power bands. Barplots for the Heat-the-Chair experiment.

**Figure 17 sensors-24-01174-f017:**
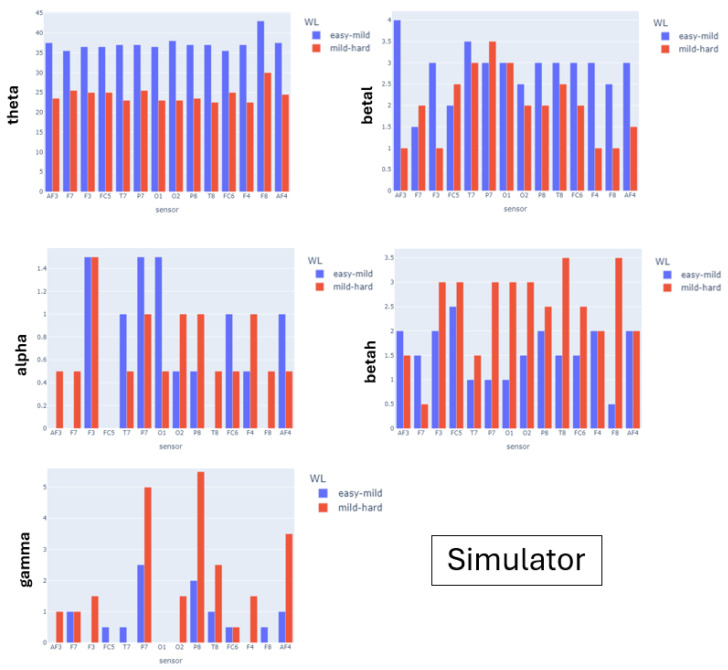
Number of peaks of EEG node power bands. Barplots for the simulator experiment.

**Figure 18 sensors-24-01174-f018:**
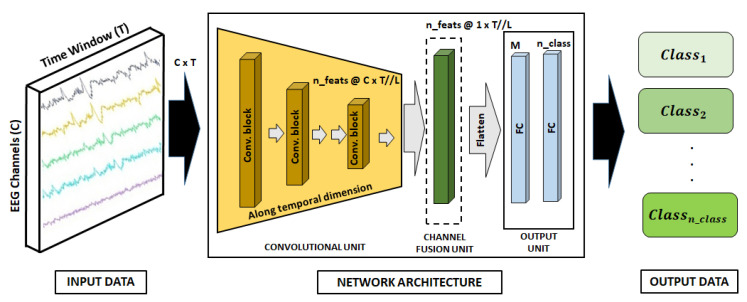
Neural network architecture for mid-fusion.

**Figure 19 sensors-24-01174-f019:**
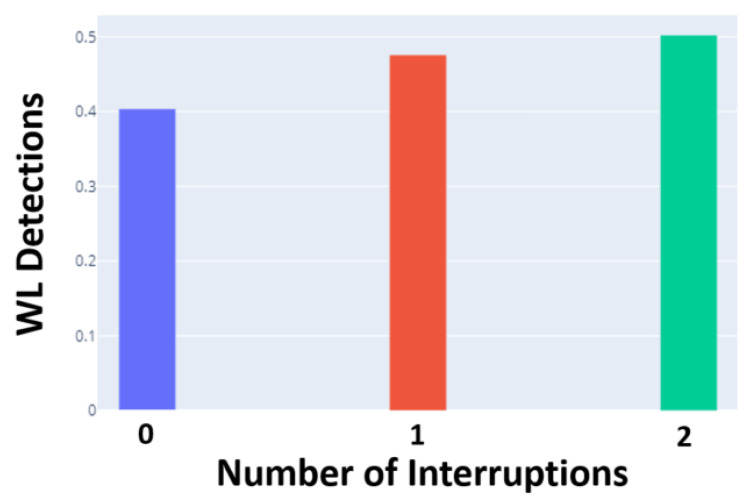
Correspondence between interruptions in the Heat-the-Chair game and the number of WL detection.

**Figure 20 sensors-24-01174-f020:**
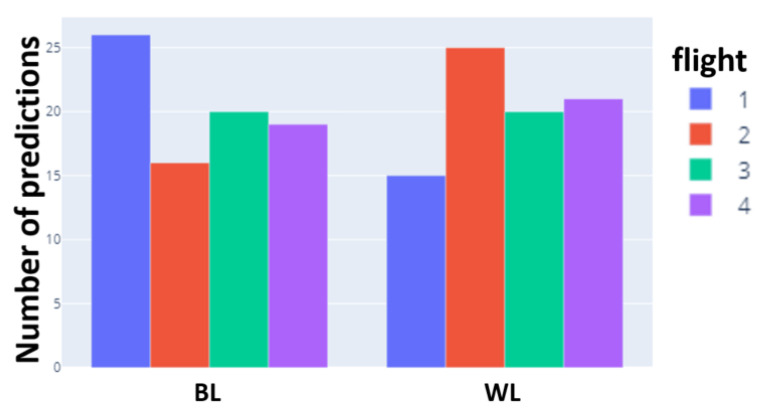
Correspondence between the number of interruptions in the Heat-the-Chair game and the number of WL detections by the neural network.

**Table 1 sensors-24-01174-t001:** Pilots information.

Pilot	Gender	Age	Flight Hours
Pilot 1	Male	51	4000
Pilot 2	Male	32	1700

**Table 2 sensors-24-01174-t002:** Pilots roles.

Experiment	Pilot = Pilot 1	Pilot = Pilot 2
	Observer = Pilot 2	Observer = Pilot 1
Easy	-	Flight 1
Medium	Flight 2	Flight 5
Medium-Hard	-	Flight 4
Hard	Flight 3	-

**Table 3 sensors-24-01174-t003:** Number of peaks of EEG node power bands. Summary of the descriptive statistics for the simulator experiment.

	Easy-Mild	Mild-Hard
theta	37.25 ± 1.73	24.39 ± 1.88
alpha	0.64 ± 0.58	0.68 ± 0.36
beta_l	2.86 ± 0.58	2.00 ± 0.80
beta_h	1.57 ± 0.52	2.46 ± 0.83
gamma	0.68 ± 0.75	1.68 ± 1.76

**Table 4 sensors-24-01174-t004:** Number of peaks of EEG node power bands. Summary of the descriptive statistics for the n_back test.

	BL	Low	Medium	Hard
theta	95.43 ± 30.52	405.93 ± 213.24	458.57 ± 63.02	632.14 ± 100.31
alpha	125.79 ± 56.91	442.93 ± 129.18	407.07 ± 63.41	623.43 ± 72.52
beta_l	101.43 ± 25.68	440.14 ± 75.51	371.64 ± 44.39	597.43 ± 83.51
beta_h	122.49 ± 42.75	524.50 ± 145.39	294.86 ± 49.20	596.29 ± 115.51
gamma	143.57 ± 49.29	553.57 ± 181.67	259.29 ± 28.64	553.86 ± 142.94

**Table 5 sensors-24-01174-t005:** Number of peaks of EEG node power bands. Summary of the descriptive statistics for the Heat-the-Chair game.

	Without	With
theta	89.50 ± 44.29	160.57 ± 41.10
alpha	75.50 ± 27.20	115.00 ± 33.34
beta low	63.43 ± 31.00	87.79 ± 25.69
beta high	51.50 ± 38.17	71.93 ± 25.21
gamma	37.79 ± 23.91	63.50 ± 30.49

**Table 6 sensors-24-01174-t006:** Number of peaks of EEG node power bands. Regression model for the N-back test.

		Coefficient	*p*-Value	95% CI
** θ **	Low	5.93 × 10^0^	-	(326.66, 425.48)
	Medium	1.89 × 10^−1^	0.02	(394.68, 514.09)
	Hard	5.07 × 10^−1^	<0.001	(542.43, 706.53)
** α **	Low	6.06 × 10^0^	-	(383.55, 471.91)
	Medium	2.81 × 10^2^	<0.001	(361.16, 452.99)
	Hard	4.98 × 10^2^	<0.001	(577.51, 669.34)
** $\beta_{l}$ **	Low	6.07 × 10^0^	-	(400.17, 468.23)
	Medium	−1.62 × 10^−1^	<0.001	(340.27, 398.14)
	Hard	3.10 × 10^−1^	<0.001	(545.68, 638.48)
** $\beta_{h}$ **	Low	6.21 × 10^0^	-	(432.24, 568.05)
	Medium	−5.42 × 10^−1^	<0.001	(251.50, 330.53)
	Hard	1.59 × 10^−1^	0.03	(506.76, 665.99)
** γ **	Low	6.25 × 10^0^	-	(434.64, 598.50)
	Medium	−6.95 × 10^−1^	<0.001	(216.88, 298.64)
	Hard	4.06 × 10^−2^	0.67	(452.64, 623.28)

**Table 7 sensors-24-01174-t007:** Number of peaks of EEG power bands. Regression model for the Heat-the-Chair game.

		Coefficient	*p*-Value	95% CI
** θ **	without	4.38 × 10^0^	-	(61.19, 97.85)
	with	6.71 × 10^−1^	<0.001	(119.76, 191.50)
** α **	without	4.24 × 10^0^	-	(54.09, 84.98)
	with	4.63 × 10^−1^	<0.001	(85.90, 134.95)
** $\beta_{l}$ **	without	3.99 × 10^0^	-	(39.07, 69.27)
	with	4.47 × 10^−1^	<0.001	(61.11, 108.35)
** $\beta_{h}$ **	without	3.60 × 10^0^	-	(20.72, 52.21)
	with	6.25 × 10^−1^	0.02	(38.71, 97.56)
** γ **	without	3.44 × 10^0^	-	(21.80, 40.69)
	with	6.31 × 10^−1^	<0.001	(40.98, 76.50)

**Table 8 sensors-24-01174-t008:** Number of peaks of EEG power bands. Regression model for the flight simulator.

		Coefficient	*p*-Value	95% CI
** θ **	easy-mild	3.62 × 10^0^	-	(35.92, 38.51)
	mild-hard	−4.25 × 10^−1^	<0.001	(23.48, 25.17)
** α **	easy-mild	6.43 × 10^−1^	-	(0.37, 0.92)
	mild-hard	3.57 × 10^−2^	0.85	(0.40, 0.95)
** $\beta_{l}$ **	easy-mild	2.86 × 10^0^	-	(2.46, 3.26)
	mild-hard	−8.57 × 10^−1^	<0.001	(1.60, 2.40)
** $\beta_{h}$ **	easy-mild	1.57 × 10^0^	-	(1.17, 1.97)
	mild-hard	8.93 × 10^−1^	<0.001	(2.07, 2.86)
** γ **	easy-mild	6.79 × 10^−1^	-	(−0.09, 1.45)
	mild-hard	1.00 × 10^0^	0.06	(0.91, 2.45)

## Data Availability

The dataset is publicly available at the Digital Document Deposit of the Universitat Autònoma de Barcelona, accessible at https://doi.org/10.5565/ddd.uab.cat/259591 (accessed on 26 December 2023). No registration is required for anyone who would like to download and use these data.

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
