# Peer review of "EEG Dataset Collection for Mental Workload Predictions in Flight-Deck Environment"

_sensors, 2024, doi:10.3390/s24041174_

Round 1
Reviewer 1 Report
Comments and Suggestions for Authors
attached

Author Response
We thank the reviewer for his/her time and advice. Attached there are our replies.

Reviewer 2 Report
Comments and Suggestions for Authors Demographic information could be more consistently presented across all groups of participants (age and gender are missing for some). In line 459, authors provide an explanation of lower correlation strength with the most difficult flight, however, references supporting their explanation are missing. Capability of task transfer (Figure 19) should be explained in greater detail. The abbreviations BL and WL are not explained, and It is not clear how the authors obtained their results, making the Figure 19 difficult to interpret. Currently used figures seem to have low resolution and are thus difficult to read. Comments on the Quality of English Language While the ideas in the manuscript are interesting and well presented, there are several areas where improvement in the English language could enhance the overall readability of the manuscript. The authors should conduct a thorough proofread to catch and correct the typographical and grammatical errors, as well as the repetitions in the manuscript.Author Response
We thank the reviewer for his/her time and advice. Attached there are our replies.

Reviewer 3 Report
Comments and Suggestions for Authors
Technically, the data presentation, method explanations, and analysis are weak. The references seem old. There are newer studies on stress analysis based on simultaneous ECG and EEG monitoring. The manuscript does not explain how they correlate EEG and ECC features. More than that, it does not even explain what ECG and EEG features (only alpha, beta, and gamma are mentioned) were used. The literature review is not sufficient, for example, what EEG and ECG features were used in relation to stress or workload in the past? There are well known features. In terms of data presentation, tables would be useful for understanding the results, not just graphs. The machine learning part lacks information on how the training was performed. What percentage of data was used for training and testing? Do the results refer to testing data only?
Typo: Line 113 and Line 11: hearth rate, change to heart rate
In correct reference Repetition: Hernández-Sabaté, A.; Yauri, J.; Folch, P.; Piera, M.À.; Gil, D. Recognition of the Mental Workloads of Pilots in the Cockpit Using 585 EEG Signals. Applied Sciences 2022, Vol. 12, Page 2298 2022, 12, 2298. https://doi.org/10.3390/APP12052298.
Comments on the Quality of English LanguageSeveral typos; the manuscript needs thorough language revision
Author Response

(The authors gave the same response as above.)

Round 2
Reviewer 1 Report
Comments and Suggestions for Authors
the article can be published in present form
Reviewer 3 Report
Comments and Suggestions for Authors
Most of the suggestions were implemented.
Comments on the Quality of English Languageaverage